# An Active Vibration Isolation and Compensation System for Improving Optical Image Quality: Modeling and Experiment

**DOI:** 10.3390/mi14071387

**Published:** 2023-07-07

**Authors:** Min Wang, Jing Xiong, Shibo Fu, Jiheng Ding, Yi Sun, Yan Peng, Shaorong Xie, Jun Luo, Huayan Pu, Shilin Shao

**Affiliations:** 1School of Mechatronic Engineering and Automation, Shanghai University, Shanghai 200444, China; xmwangmin@shu.edu.cn (M.W.); lemonbear@shu.edu.cn (J.X.); fushibo@shu.edu.cn (S.F.); ding_jiheng@shu.edu.cn (J.D.); yisun@shu.edu.cn (Y.S.); pengyan@shu.edu.cn (Y.P.); phygood_2001@shu.edu.cn (H.P.); 2Engineering Research Center of Unmanned Intelligent Marine Equipment, Ministry of Education, 99 Shangda Rd., Shanghai 200444, China; srxie@shu.edu.cn; 3School of Computer Engineering and Science, Shanghai University, Shanghai 200444, China; shussl@shu.edu.cn; 4The State Key Lab of Mechanical Transmission, Chongqing University, Chongqing 400044, China

**Keywords:** vibration isolation, optical detection equipment, evaluation metric

## Abstract

Optical detection equipment (ODE) is subjected to vibrations that hamper the quality of imaging. In this paper, an active vibration isolation and compensation system (VICS) for the ODE is developed and systematically studied to improve the optical imaging quality. An active vibration isolator for cameras is designed, employing a dual-loop control strategy with position compensation and integral force feedback (IFF) control, and establishing the mapping relationship between vibration and image quality. A performance metric for evaluating images is also proposed. Finally, an experimental platform is constructed to verify its effectiveness. Based on the experimental results, it can be concluded that the proposed VICS effectively isolates vibrations, resulting in a reduction of 13.95 dB in the peak at the natural frequency and an 11.76 Hz widening of the isolation bandwidth compared with the system without it. At the same time, the experiments demonstrate that the image performance metric value increases by 46.03% near the natural frequency.

## 1. Introduction

With the rapid development of unmanned driving and autonomous navigation, the utilization of optical detection equipment (ODE) has expanded across various fields [1,2,3,4,5]. The requirements of the working environment for the ODE are quite stringent and can effectively prevent reduced measurement accuracy due to vibrations [6,7]. Consequently, research on vibration suppression in the ODE is currently garnering the attention of numerous scholars.

To reduce the influence of unstable factors such as vibration during the optical imaging process, some scholars use digital image enhancement methods to improve image quality using computer compensation algorithms in the later stages [8,9,10,11]. This generates a large volume of data accumulation during post-processing, which affects real-time performance. This disadvantage also greatly reduces the adaptability of the ODE. Introducing a vibration isolation and compensation system (VICS) between the ODE and vibration foundation has been proven to be effective in improving image quality [12,13,14]. The key to determining the performance of VICS in the ODE is to focus on three aspects: a VICS, an optical stabilization control algorithm, and an evaluation of image performance related to vibration.

VICSs for improving the image quality of the ODE are studied. Li et al. [15] compared the vibration reduction effect of different isolation dampers on cameras onboard unmanned aerial vehicles. Lin et al. [16] studied a double-layer VICS using a four-point support symmetric radiation arrangement, which can reduce image motion to less than 0.1 pixels. Verma et al. [17] designed a Stewart platform to reduce the vibration of drone image sensors. Shao et al. [2] proposed an active suspension mechanical system to achieve stable image acquisition for cameras mounted on moving vehicles. The effectiveness of adding an isolation mechanism in reducing the impact of vibration on image quality is undoubtedly evident, but more research seems less widespread.

Regardless of the ODE, extensive research is being conducted on VICSs [18,19,20,21]. Shin et al. [22] investigated a three-degree-of-freedom vibration isolator for unmanned aerial vehicle detection systems and reduced the isolation performance in three directions to 98.3%, 94.0%, and 94.5% in the range of 30–85 Hz. Jiang et al. [23] employed quadrilateral mechanisms and lateral springs as a single layer, with a long rod as the connection between each layer, to achieve a better vibration reduction performance with a smaller device than other vibration isolators. Yan et al. [24] designed a biomimetic toe mechanism inspired by the movements of felines and demonstrated that this mechanism has a wide displacement range and performs well in low-frequency vibration reduction. Therefore, VICS configurations in other fields can provide more new ideas for the ODE.

VICSs can be used to reduce the effects of vibrations on image quality. However, there is still a difficulty that has not been solved. As the isolation performance is limited, a passive vibration isolation system cannot be applied to higher-demand scenarios [25,26]. Active vibration control can effectively compensate for the shortcomings of passive vibration isolation systems with the use of control algorithms [27,28,29,30]. Chang et al. [31] designed an active control technique with visual feedback using adaptive sliding mode control and filtered LMS algorithms for vibration suppression comparison and verified the effectiveness of both visual feedback-based active control methods in suppressing low-frequency vibrations of equipment on translation or pitch platforms. Zhao et al. [32] employed a control method combining a linear quadratic regulator with an adaptive neural network to control a 1/4 suspension system. Negash et al. [33] and Cheng et al. [34] researched a feedback control strategy based on a skyhook, with peak values of 67.83% and 15.2%, respectively. To improve the image quality of the ODE, some scholars have found that this method is also applicable. Zhao [35] et al. studied a variable rigid shock absorber using reinforcement learning optimization stiffness control for remote sensing image satellites to improve image quality. Sun et al. [36] proposed a bridge-type mechanism with a piezoelectric actuator for stabilizing the images of infrared imaging systems, and the image stabilization performance of the designed mechanism was verified through image sequence control and feedback signal processing. In summary, image quality can indeed be raised by additional optical stabilization control algorithms on the basis of passive VICSs.

Vibration inevitably affects the image quality captured by the ODE [8,37]. As the amplitude of vibration increases, the image will become blurry [8,17]. Lee et al. [38] found that the image quality of unmanned aerial vehicles decreases in image quality due to vibration during a bridge inspection. However, the quantitative relationship between the amplitude of vibration and image quality is unclear. For the image performance evaluation related to vibration, Mohamed et al. [39] studied the impact of satellite micro-vibration on satellite image quality and proposed an algorithm to simulate micro-vibration. Suresh et al. [40] proposed an approach for the performance evaluation of complex image processing algorithms. The performance of the proposed and existing algorithms was compared using evaluation metrics. Some scholars [41,42,43] evaluated vibration through the frequency- and time-domain analyses of the response, with evaluation metrics including natural frequency values, peak natural frequency values, and isolation bandwidths. Negash et al. [33] used the root-mean-square (RMS) value of the vertical displacement of a loaded mass as the evaluation criterion. After analyzing the literature, we found that a clearer mapping relationship between the vibration amplitude and image quality has not yet been established.

Based on these points, in this paper, a new indicator is proposed to explore the relationship between image quality and vibration, and a position compensation loop is proposed to counteract the impact of the vertical position of the ODE. Compared with traditional indicators, the relationship between image quality and vibration amplitude is more intuitively reflected in the IPEI. To better validate the IPEI, an active VICS for cameras is designed to ensure the accurate alignment of the active actuation force of the voice coil motor, utilizing tandem flexible hinges and membrane springs. An active vibration algorithm is applied to the feedback control loop, working alongside the position compensation loop as a dual-loop control strategy. The proposed system decreases the peak at the camera’s inherent frequency and broadens the isolation bandwidth. Based on image performance metrics, the proposed vibration isolation system can improve image quality.

The organization of this paper is as follows: In Section 2, the design principles for the performance metric are outlined. In Section 3, the structure of the vibration isolation device is proposed. In Section 4, the theoretical model is established while also considering the weight impact of the ODE. The dual-loop control strategy is also described in this section. In Section 5, the results of the experimental tests conducted for the verification of the vibration isolation performance of the system and the validity of the proposed performance metric are presented. Finally, the main conclusions are drawn in Section 6.

## 2. Design of the Image Performance Evaluation Index (IPEI)

In this section, we propose the design of the image performance evaluation index. This metric mainly used to demonstrate that the image quality is affected by the amplitude of transmissibility in vibration evaluation. Typically, the data collected using the ODE are stored in the form of video footage, where the video continuously displays the sequence of images. When the number of images viewed within a specific time interval exceeds a certain threshold, the human eye perceives the image as being in motion. We can characterize the motion of an object by examining the displacement of the image at identical positions because the most prominent manifestation of vibration is the amplitude of the object’s motion, which can be reflected through two consecutive image frames.

A pixel point in Image I is assumed as follows:(1)$$c={\;\left(B_{h,w},G_{h,w},R_{h,w}\right)}_{I}$$
where *B*, *G*, and *R* represent the grey values of blue, green, and red, respectively, and h and w are the pixel positions corresponding to the rows and columns, respectively. $N\left(c\right)$ is the neighborhood of the pixel point. In the subsequent frame image *K*, the neighborhood $N\left(c,\delta \right)$ exists in ${\left(h^{\prime },w^{\prime }\right)}_{K}$, resulting in the following relationship:(2)$$\left\{\begin{array}{c} B_{h,w}=B_{h^{\prime },w^{\prime }}\\ G_{h,w}=G_{h^{\prime },w^{\prime }}\\ R_{h,w}=R_{h^{\prime },w^{\prime }} \end{array}\right.$$

The pixel point is $c^{\prime }={\left(B_{h^{\prime },w^{\prime }},G_{h^{\prime },w^{\prime }},R_{h^{\prime },w^{\prime }}\right)}_{K}$, and Pixel Point c and Pixel Point $c^{\prime }$ represent the same point. In this case, $cc^{\prime }$ is the moving distance of the digitized image. The imaging quality is affected by the distortion between the real image and the ideal image, and the relative coordinates of the camera. Assuming the distortion coefficient *K* and the parameter matrix *M*, there exists a conversion relationship between the real image and the digitized image: $f\left(K,M\right)$. The actual image movement distance can be expressed as follows:(3)$$Y=\frac{1}{h*w}\sum_{i=1,j=1}^{h,w}{\left\|f^{-1}\left(K,M\right)\cdot \left(h,w\right)-f^{-1}\left(K,M\right)\cdot \left(h^{\prime },w^{\prime }\right)\right\|}_{2}=\beta \cdot Y_{unit},$$
where β represents the distance scale factor, and $Y_{unit}$ is the unit distance.

As shown in Figure 1, regarding Vibration Cases 1 and 2, the actual image moving distances are $Y_{1}$ and $Y_{2}$, respectively. The following relationship exists:(4)$$\left\{\begin{array}{c} \frac{Y_{1}}{Y_{unit}}=\beta_{1}\\ \frac{Y_{2}}{Y_{unit}}=\beta_{2} \end{array}\right.,$$

Equation (4) can be modified as follows:(5)$$\frac{Y_{1}}{Y_{2}}=\frac{\beta_{1}}{\beta_{2}},\;$$

When $\frac{\beta_{1}}{\beta_{2}}>1$, the vibration amplitude in Case 1 is greater than that in Case 2. When $\frac{\beta_{1}}{\beta_{2}}<1$, the vibration amplitude in Case 1 is smaller than that in Case 2. Therefore, we can conclude that there is a positive correlation between vibrations and image displacements.

In practical situations, directly obtaining image displacements is not easy. However, it is observed that as the image displacement per unit of time increases, the image becomes blurrier (lower image quality). External environmental factors also have an impact on image quality [6]. Their mapping relationship is as follows:(6)$$\Psi \left(y\right)=\Phi \left\{\phi \left(Y\right),\varphi \left(T,U,w\right)\right\},$$
(7)$$\Psi \left(y\right)\propto \frac{1}{\phi \left(Y\right)},$$
where Ψ is the image quality; y is the pixel point’s moving distance; ϕ is the influence of vibration; Y is the actual image moving distance, which is the amplitude of the vibration; φ is the influence of environmental factors; T is the ambient temperature; U is the ambient humidity; and w is the ambient wind speed. From the aforementioned analysis, it can be deduced that the magnitude of vibration is negatively correlated with image quality. Therefore, we propose the following image performance metric:(8)$$P_{m}=10\times {log}_{10}\left(\frac{{\left(2^{n}-1\right)}^{2}}{\frac{1}{n}\sum_{i=1}^{n}{\left(I_{n}-K_{n}\right)}^{2}}\right),$$
where *n* represents the total number of pixel points, given by n=h×w, where h and w are the dimensions of the image. $I_{n}$ is the pixel grey value of Image *I*, while $K_{n}$ corresponds to the pixel grey value of Image *K*.

The video frame rate is set at 25 frames per second (fps), which results in approximately 1500 frames for a one-minute video. Due to the substantial number of frames acquired using the ODE, it is essential to compute the average pixel displacement between each pair of adjacent frames:(9)$$P=\frac{1}{m}\sum_{i=1}^{m}P_{m},$$
(10)$$\Psi \left(y\right)\propto P,$$
where *m* is the total number of frames.

We introduce *P* as a performance metric to assess image quality. As indicated by Equation (10), a higher value of *p* signifies better image quality, while a lower value of *p* demonstrates poorer image quality. The performance improvement can be expressed by the following formula:(11)$$Performance\;improvement=\frac{Improved\;IPEI-Original\;IPEI}{Original\;IPEI}\cdot 100\%,$$

## 3. Design of an Active VICS for the ODE

In this paper, we design a VICS using the HIKVCSION camera. The product parameters of the HIKVCSION camera are provided in Table 1.

As shown in Figure 2a, the ODE is installed on the installation platform. To minimize the impact of vibrations, it is necessary to install an active VICS between the installation platform and the camera. Traditional VICSs typically consist of a mass–spring–damper system, which has a narrow vibration isolation bandwidth and is not effective in suppressing vibrations in the low-frequency range. Therefore, it is necessary to introduce an active VICS. Voice coil motors (VCMs) are widely used as motors for providing active vibration isolation. Their output power is controlled by modulating the current passing through the coil, and they offer several advantages such as fast response, moderate stroke, and wide control bandwidth.

Based on the above research, we designed an active VICS based on a VCM, as shown in Figure 2b. The active VICS comprises several essential components: a load platform, a base platform, an active mechanism, a passive mechanism, two acceleration sensors, and a controller.

The load platform is used to support the ODE (such as the camera), while the base platform is fixed to the installation platform. The passive mechanism consists of spring components and connection components, with the springs ensuring the load-bearing capacity of the isolation system and acting as passive isolation elements. The active mechanism includes a VCM, a diaphragm spring, and a flexible hinge. With the low radial stiffness of the flexible hinge and the high radial stiffness of the diaphragm spring, the VCM ensures co-axiality, and the installation is easy. The low radial stiffness of the flexible hinge and the high radial stiffness of the diaphragm spring ensure that the force on the output shaft of the VCM is always aligned with the load platform, compensating for any eccentricity introduced by the load platform without losing the transmission of axial motion. The high axial stiffness of the flexible hinge and the low axial stiffness of the diaphragm spring ensure that the excitation force generated by the VCM is transmitted axially, effectively suppressing vibrations caused by excitation components in the vertical direction while compensating for displacements in other directions. An acceleration sensor is installed on the load platform to collect residual vibration signals, which are transmitted to the controller. After processing using the built-in feedback algorithm, the controller outputs control signals to the VCM to drive it to suppress vibrations. Another acceleration sensor is installed on the base platform to collect ground vibration signals. The vibration reduction performance of the designed VICS can be evaluated by processing the data collected from these two sensors.

## 4. Modeling and Active Control Algorithm Strategies

### 4.1. Modeling of the VICS for the ODE

The classical simplified schematic diagram of the vibration isolator mechanism is presented in Figure 3. Elastic components such as the coil spring, diaphragm spring, and flexible hinge are equivalent to a stiffness element with a stiffness of *K*. The equivalent damping of the system is *C*. The load platform is equivalent to a mass block with a mass of M, and the base platform is equivalent to a mass block with a mass of m. The VICS we designed is a single-degree-of-freedom system. We modeled it based on Newton’s Second Law to obtain an analytical solution for its vibration transmission rate. The simplified isolation mechanism has a dynamic model:(12)$$\left\{\begin{array}{c} M{\ddot{x}}_{1}+K_{e}\left(x_{1}-x_{0}\right)+C\left({\dot{x}}_{1}-{\dot{x}}_{0}\right)+u=0\\ {u=F}_{fb} \end{array}\right.,$$
where *M* is the mass of the load platform, $K_{e}$ is the equivalent stiffness of the system, *C* is the equivalent damping between the load platform and the base platform, $x_{1}$ is the vibration displacement of the load platform, $x_{0}$ is the vibration displacement of the base platform, and $F_{fb}$ is the feedback output force from the VCM. The M, $K_{e}$, and *C* are calibrated using system identification methods, as shown in Table 2.

The impact of camera mass cannot be ignored since the camera mass and load platform mass have similar orders of magnitude. To ensure that the vertical position of the ODE is not affected by the VICS, we studied the relationship between the static displacement of the ODE and the system stiffness in the vertical direction. The VICS is simplified, as shown in Figure 4. The load platform is simplified as a flat plate with a mass of *M*. The coil and diaphragm springs are connected in parallel and therefore are equivalent to a spring with a length of h, and their equivalent stiffness is as follows:(13)$$K_{e}=K_{C}{+K}_{D},$$
where $K_{C}$ is the stiffness of the coil spring, and $K_{D}$ is the stiffness of the diaphragm spring. The installation platform is rigidly connected to the base platform, which is simplified as a flat plate with a mass of m. Figure 4b is the schematic diagram of the VICS when there is a load platform. G is the geometric center of the position of a hypothetical camera with no weight. By analyzing the forces acting on it, the following equation can be obtained:(14)$$M\cdot g=K_{e}\cdot \left(h-z\right),\;$$
where *g* is the local gravity acceleration, taken as $g=9.8\;N/m^{2}$; *h* represents the initial length of the spring, while *z* is the length of the spring after being subjected to the gravitational force of the load platform, which is determined as follows:(15)$$z=h-\frac{M\cdot g}{K_{e}},\;$$

Figure 4c illustrates the schematic diagram of the VICS with the load platform and camera. By conducting a force analysis on it, we can obtain the following equation:(16)$$\left(M+∆m\right)\cdot g=K_{e}\cdot \left[h-\left(z-\xi \right)\right],$$
where ξ represents the displacement variation in the vertical direction of the camera’s center. It can be expressed using the following equation:(17)$$\xi =\frac{∆m\cdot g}{K_{e}}.\;$$

Thus, the relationship between the vertical static displacement of the ODE and the system stiffness is given as follows:(18)$$\xi =\frac{∆m\cdot g}{K_{C}{+K}_{D}}.\;$$

Consequently, Equation (12) can be rewritten as follows:(19)$$\left\{\begin{array}{c} \left(M+∆m\right){\ddot{x}}_{1}+K\left(x_{1}-x_{0}\right)+C\left({\dot{x}}_{1}-{\dot{x}}_{0}\right)+u=0\\ {u=F}_{fb}+F_{pc} \end{array}\right.,$$
where $F_{pc}$ is the position compensation force output by the VCM.

### 4.2. Dual-Loop Control Strategy of the ODE

In this section, the control strategy developed for the proposed VICS of the ODE is presented. The IPEI will not really affect the active effect as an evaluation system. Figure 5 illustrates the flowchart of our control strategy, which comprises two independent control loops: the position compensation loop and the feedback vibration control loop. The position compensation loop serves to counteract the impact of the vibration isolation platform on the vertical position of the ODE. In accordance with the preset parameters, the position compensation controller is used to calculate the position compensation control signal. After filtering the noise signal, the VCM output is adjusted by the position compensation control transfer function, enabling the ODE to achieve position compensation at the designated location. The feedback control loop is employed to suppress vibrations. Within the feedback control loop, the feedback acquisition sensor gathers the load platform’s vibration signal. After noise filtering, the control parameters are computed using the feedback controller, and the parameters are added. Following the passage of the control signal, which is calculated using feedback control, through the filter, the residual vibration after control is obtained via the control transfer function. This residual vibration acts on the load platform in conjunction with the vibrations following open-loop transmission and the vibrations after position compensation control, thus completing a comprehensive control set. The control strategy of each loop is detailed below.

In the previous section, we derived a relationship between the static displacement variation in the vertical direction of the ODE and the system stiffness. We propose a method of position compensation for the ODE to compensate for static displacement. In our vibration isolation strategy for optical detection systems, active vibration reduction is primarily accomplished using a VCM. The operating principle of the VCM relies on the Ampere force principle, whereby an energized conductor produces a force F within a magnetic field, as expressed using the following formula:(20)F=nBLI=αI,
where B is the magnetic induction intensity, I is the electric current flowing through the conductor, and L is the wire length. There is a linear positive correlation between *F* and *I*, while α signifies the output coefficient of the VCM. Consequently, the VCM output can be controlled by adjusting the magnitude of I. Based on Equations (18) and (20), the current bias can be obtained as follows:(21)$$∆I=-\frac{∆m\cdot g}{\alpha },$$
where ∆I is the current bias, ∆m is the camera mass, and *g* is the local gravitational acceleration. The current required to control the VCM can be expressed as follows:(22)$$I_{a}=∆I+I_{fb},$$
where $I_{a}$ is the current that controls the VCM, ∆I is the current bias, and $I_{fb}$ is the current calculated using the feedback control algorithm. The position compensation force output of the VCM can be expressed as follows:(23)$$F_{pc}=\alpha \cdot ∆I.\;$$

In the design of the vibration isolation scheme presented in this paper, we consider the issues of high natural frequency and narrow vibration isolation frequency bands inherent in traditional passive mechanisms. As a result, we developed an active control algorithm to lower the natural frequency and broaden the vibration isolation frequency band. The skyhook control algorithm is commonly used in engineering, and the integral force feedback (IFF) control algorithm is a special form of the skyhook control algorithm. The IFF can alter the system’s damping through the integral component, creating a skyhook damping effect, which ensures high-frequency attenuation rates while suppressing the amplitude of vibrations at the resonance peak, achieving a skyhook damping effect. Therefore, we designed an IFF algorithm for the vibration isolation scheme. The control of the IFF can be expressed as follows:(24)$$F_{fb}\left(s\right)=F_{IFF}\left(s\right)=\frac{1}{s}\cdot \left(M\cdot \gamma \right)\cdot a_{feedback}\left(s\right)=\frac{1}{s}\cdot \left(M\cdot \gamma \right)\cdot s^{2},\;$$
where $k_{I}$ = M · γ is the integral gain coefficient, and $a_{feedback}$ is the acceleration sensor reading.

The transfer function of the IFF control algorithm adopted in this paper can be expressed as follows:(25)$$G_{IFF}\left(s\right)=\frac{Cs+k_{e}}{\;M\;\cdot s^{2}+\left(C+k_{I}\right)\cdot s+k_{e}\;}\approx \frac{1}{\frac{1}{\omega_{n}^{2}}\cdot s^{2}+\frac{\gamma }{\omega_{n}^{2}}\;\cdot s+1},\;$$
where the system is a vibration system without equivalent damping when γ≫C, and $\omega_{n}$ is the natural frequency of the system. The stability of the IFF control used is mainly caused by time delay. When $k_{I}$ values are very radical, time delay is increased. Therefore, to maintain the stability of the system, the IFF gain parameters should be taken within the limit value.

A schematic representation of the feedback control loop strategy is depicted in Figure 6a. The external excitation signal, after passing through the vibration isolation platform, is collected using the feedback sensor as a feedback signal, which is then filtered and output to the controller. The controller subsequently outputs the feedback control signal to the driver, driving the VCM to counteract the vibration and achieve vibration control. As illustrated in Figure 6b, $X\left(s\right)$ is the external excitation, $G_{1}\left(s\right)$ is the transfer function of the external excitation $X\left(s\right)$ transmitted to the base platform, $D\left(s\right)$ signifies the response of the external excitation $X\left(s\right)$ transmitted to the base platform, and $E_{1}\left(s\right)$ indicates the difference between the response transmitted to the base platform using the external excitation $X\left(s\right)$ and the feedback control force, which corresponds to the residual vibration after feedback control. $G_{2}\left(s\right)$ is the open-loop transfer function of the isolation device, and the response of the residual vibration $E_{1}\left(s\right)$ after passing through the isolation device is $Y\left(s\right)$. The feedback collection sensor collects the signal $G_{2}\left(s\right)$ and transmits it to the feedback controller $P\left(s\right)$, which outputs the feedback control force $F_{fb}\left(z\right)$. The calculation formula is as follows:(26)$$\left[X\left(s\right)G_{1}\left(s\right)-Y\left(s-1\right)P\left(s\right)\right]G_{2}\left(s\right)=Y\left(s\right)$$

## 5. Experiment

### 5.1. Experimental Platform Construction

On the basis of the above research, we manufactured an active VICS, with detailed materials shown in Table 3. To verify the effectiveness of the position compensation loop, we built an experimental platform and conducted experiments on it. The position compensation was achieved using the current bias, and the conversion relationship between the two in practical applications is shown in Figure 7 (the model of the experimental equipment is also shown). In the position compensation loop, the load quality (camera quality) was sent to a computer and then transmitted to the PXI controller. The PXI controller was used to calculate the position compensation control signal, which is an analog voltage signal. This signal was sent to the driver of the VCM, and the driver outputted a current signal to drive the VCM to generate a position compensation control force. The output coefficient of the VCM used was 28 N/A, the output coefficient of the VCM diver was 0.077 A/V, and the camera quality was 650 g. Using Equation (21), the analytical solution of the current ∆I was obtained as 0.227 A, and the voltage was 2.95 V. As shown in Figure 8, noncontact laser displacement sensors were used to measure the distance between the load platform and the sensor under three different vibration isolation device states. The experimental results are shown in Figure 9. From 0 to 5 s, no camera was placed, while from 5 to 10 s and from 10 to 15 s, the position compensation control force was applied. If the initial position was 0 when the camera was not placed, starting from 5 s, due to the placement of the camera, the load platform would move down. At 10 s, the control was turned on, and the load platform returned to the initial position. This experiment demonstrated the effectiveness of the position compensation loop well.

### 5.2. Experiment of the Vibration Isolation Performance

Next, we set up experiments to test the vibration isolation and compensation performance. As shown in Figure 10, the real-time active control system consisted of an acceleration sensor, a charge amplifier, a controller, a driver, and the VICS. The spectrum test and analysis system consisted of two acceleration sensors, a power amplifier, a vibration exciter, and a data acquisition system. The camera was mounted on the load platform, and the base platform was placed on the vibration exciter. Two acceleration sensors were installed on the load platform and the exciter platform and were connected to a sensor signal conditioner. The vibration exciter was driven by inputting a control signal to a power amplifier via a spectrum analyzer (data acquisition system) to generate an excitation signal to simulate interference with the camera. When the vibration exciter operated, the vibration signal passed through the vibration isolator and reached the camera. The acceleration sensor on the load platform collected the signal, which represented the acceleration of the camera and was also the feedback signal. The signal was modulated using a sensor signal conditioner and then inputted to a junction block of the controller. The feedback algorithm built into the controller calculated the feedback control signal, which was sent to the driver of the VCM to produce the feedback control force. The base platform was connected to the vibration exciter, and the base sensor collected the ground disturbance acceleration. The acceleration signals of the base and load platform were inputted to the spectrum analyzer (data acquisition system) to obtain the system transfer function for analyzing the vibration attenuation situation. The upper computer could monitor and process the data in real time, including the image captured using the camera and the vibration transfer function curve. Table 4 lists the model numbers of the experimental equipment.

To verify the effectiveness of the designed VICS, we conducted two sets of experiments (fixed connected camera and fixed connected camera with VICS), using white noise to excite the device and using acceleration sensors to measure acceleration. The data were processed in LMS software.

Figure 11 presents the transmissibility curves under two different conditions: one condition in which the camera was directly mounted on the installation platform (red line) and another condition in which a designed VICS was placed between the camera and the installation platform (blue line).

Regarding the red line, the camera and installation platform cannot be considered rigid bodies because there is no true fixed connection. The red line is the line oscillating near 0 dB. Regarding the blue line, the natural frequency of the vibration isolation system under passive control is 13.67 Hz, and the resonance peak can reach 12.11 dB. In the figure, the vibration isolation performance is approximately equivalent at 19.25 Hz, 39.65 Hz, and 42.75 Hz for both conditions. The performance is better when the curve is low but worse when the curve is high. In the blue regions (I, III) of the figure, adding the passive VICS is counterproductive for vibration suppression, while in the red regions (II, IV), it is beneficial. However, the areas of Regions I and III are much smaller than the combined areas of Regions II and IV, indicating that the addition of the isolator reduces the isolation performance in the blue regions but optimizes it in the red regions (which are larger).

To expand the advantage at a natural frequency and further compensate for the isolation performance at the isolation bandwidth, we introduced an active control algorithm to the VICS. In previous research work, we found that applying the IFF control suppresses the peaks at natural frequencies, which are mostly where reduced isolator performance occurs. Therefore, we added the IFF control to reduce the isolator performance of natural frequencies.

As shown in Figure 12, we compared the transmissibility curves under three different conditions: the red line represents the case in which the camera was rigidly attached to the installation platform, the blue line represents the case with the addition of a designed VICS between the camera and the installation platform, and the green line represents the case with the addition of the IFF control.

In the blue regions (I, III) of the figure, adding the IFF control to the VICS contributes to vibration suppression, while in the red regions (II, IV), it is beneficial. However, the areas of Regions I and III are much smaller than the combined areas of Regions II and IV, suggesting that the addition of the VICS reduces the isolation performance in the blue regions but significantly improves it in the red regions. Moreover, the area of the blue regions in Figure 12 is much smaller than that in Figure 11, indicating that the isolation performance reduction achieved with the addition of the IFF control algorithm is less than that achieved with the passive control strategy. After incorporating the IFF control, the peak at the natural frequency of 13.67 Hz decreased from 12.11 dB to −1.84 dB, resulting in a 13.95 dB attenuation. This demonstrates that the IFF control had a significant vibration reduction effect on the system and could effectively suppress the resonance phenomenon under a natural frequency. The system’s initial isolation frequency decreased from 19.25 Hz to 7.49 Hz. Furthermore, the isolation bandwidth widened by 11.76 Hz, which improves the vibration isolation performance.

Therefore, by introducing the IFF control algorithm for the VICS, we reduced the areas in which the isolation performance deteriorated to some extent and achieved superior vibration suppression effects.

As shown in Figure 13a, the blue line represents the transmissibility curve of the passive system, while the green line signifies the transmissibility curve of the active system with the incorporated IFF control. Figure 13b–e show the system’s acceleration response under single-frequency excitation at 10 Hz, 14 Hz, 20 Hz, and 60 Hz, respectively. The blue line represents the acceleration response of the load platform for the passive system, while the green line indicates the acceleration response of the load platform for the active system with the incorporated IFF control. When the excitation frequency was near the system’s natural frequency (i.e., at 14 Hz), the amplitude attenuation was the greatest among the four frequency points, reaching 0.300 g. The specific experimental data are recorded in Table 5. These results also indicate that the IFF control could effectively suppress the system’s resonance phenomenon, thereby reducing the system’s vibration amplitude and enhancing its stability and reliability.

Figure 14b–e display the partial image data captured using the camera under 10 Hz, 14 Hz, 20 Hz, and 60 Hz single-frequency excitations as well as white noise excitation, respectively. The first three images were captured under passive conditions, while the latter three images were captured after adding the IFF control. The image quality was better when control was applied. Using our designed performance metric P, we processed the collected image data. Figure 14a displays the scatter plot of the image data collected using our designed performance metric P, with blue points representing data under passive conditions and green points representing data after adding the IFF control. In passive conditions, P was the smallest at 14 Hz, which, according to our assumption, indicates that the image data quality collected under 14 Hz single-frequency excitation was relatively poor. From previous experiments, we know that 14 Hz is near the resonance peak, i.e., the frequency point with the largest vibration. Thus, we concluded that the smaller the performance metric P was, the larger the vibration, which is consistent with our earlier assumption. When the IFF control was added, the value of P increased with the frequency, improving the image quality. We can also infer from Figure 13 that the vibration decreased as the frequency increased, indicating that the larger the performance metric was, the smaller the vibration. This suggests that our designed performance metric P can effectively reflect the system’s vibration status, thereby evaluating the system’s vibration reduction effect. The calculated results of the performance metric P are listed in Table 6. Under open-loop control with white noise excitation, the image quality’s *p* value was 23.36274. After applying the IFF feedback algorithm, the *p* value increased to 25.53195. This also proves the effectiveness of the IFF control algorithm in reducing the impact of vibrations on image quality.

## 6. Conclusions

In this paper, we proposed an active VICS for the ODE. The isolation performance was validated using a real-time active control system, and the transmissibility curve was obtained using a spectrum test and analysis system. A dynamic model was established using Newton’s method, and the analytical solution of the isolator’s transfer rate was obtained. Two control loops were designed: the position compensation control loop was used to counteract the influence of the vibration isolation platform on the vertical position of the ODE during operation, while the feedback control loop was used for vibration suppression. An experimental prototype was built, and the accuracy of the test verification model and isolation performance evaluation were examined. The experimental results showed that the vibration isolation bandwidth starting frequency decreased from 19.25 Hz to 7.49 Hz, the peak at the natural frequency decreased from 12.11 dB to −1.84 dB, and the image performance metric value increased by 46.03% near the natural frequency. Through experimental case studies, the effectiveness of the proposed VICS for the ODE was confirmed.

Additionally, we proposed a performance metric P, which quantifies the impact of vibrations on image quality through the mapping relationship between the image quality collected using the image acquisition system and the vibration, thereby evaluating the system’s vibration reduction effect. In the constructed prototype test, the smaller the value of the performance metric P was, the worse the image quality and the larger the vibration; the larger the value of the performance metric P was, the better the image quality and the smaller the vibration. This is consistent with our hypothesis, indicating that our designed performance metric P can effectively reflect the system’s vibration status, thereby evaluating the system’s vibration reduction effect. In the future, we will further optimize the design of the performance metric to better characterize the system’s vibration status.

## Figures and Tables

**Figure 1 micromachines-14-01387-f001:**
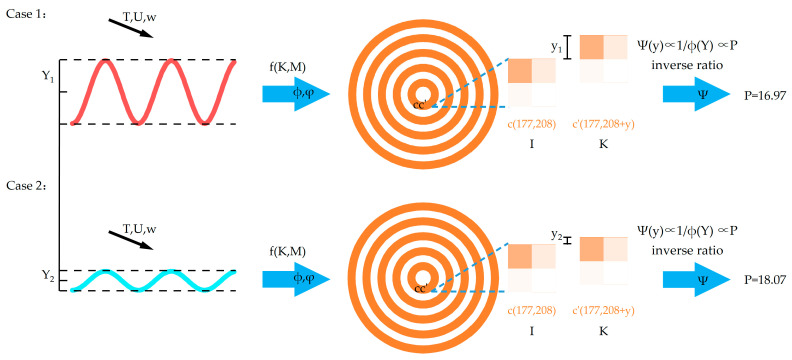
Schematic diagram of the performance index.

**Figure 2 micromachines-14-01387-f002:**
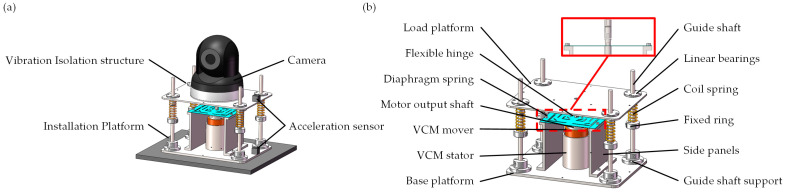
Vibration isolation device mechanism diagram: (**a**) schematic diagram of the device installation with the vibration isolation; (**b**) three-dimensional diagram of the vibration isolator.

**Figure 3 micromachines-14-01387-f003:**
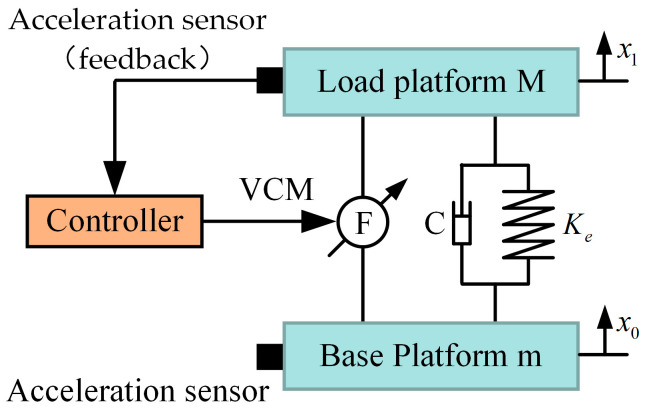
Simplified schematic diagram of the vibration isolator mechanism.

**Figure 4 micromachines-14-01387-f004:**
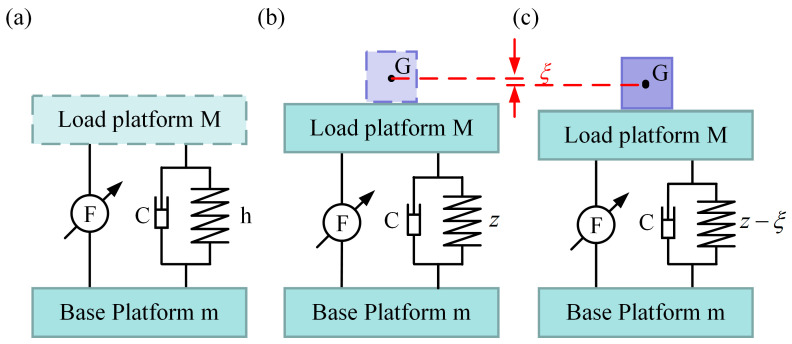
Vibration isolation device mechanism diagram: (**a**) schematic without a load platform; (**b**) schematic with a load platform; (**c**) schematic with a load platform and optical detection equipment (ODE).

**Figure 5 micromachines-14-01387-f005:**
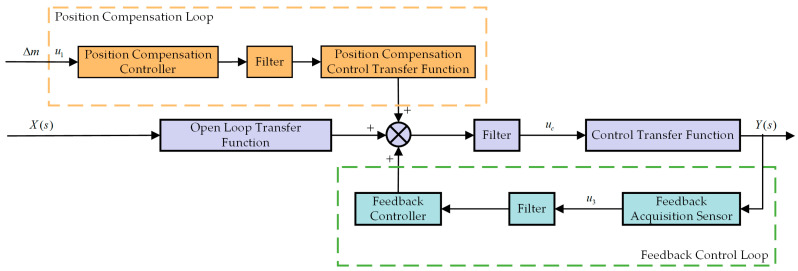
Flowchart of the control strategy for the design of the ODE vibration isolation scheme.

**Figure 6 micromachines-14-01387-f006:**
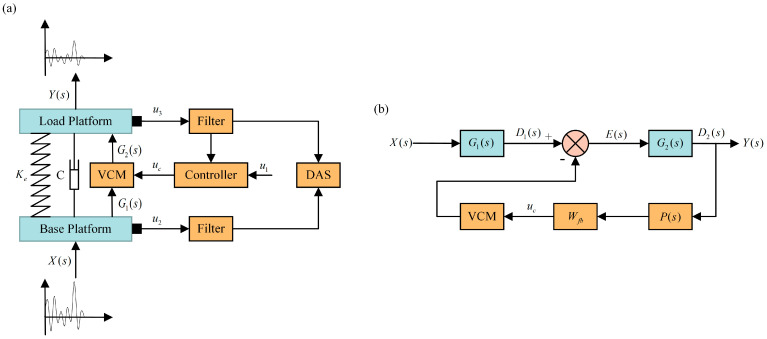
Feedback control loop control strategies: (**a**) flow diagram; (**b**) control block diagram.

**Figure 7 micromachines-14-01387-f007:**
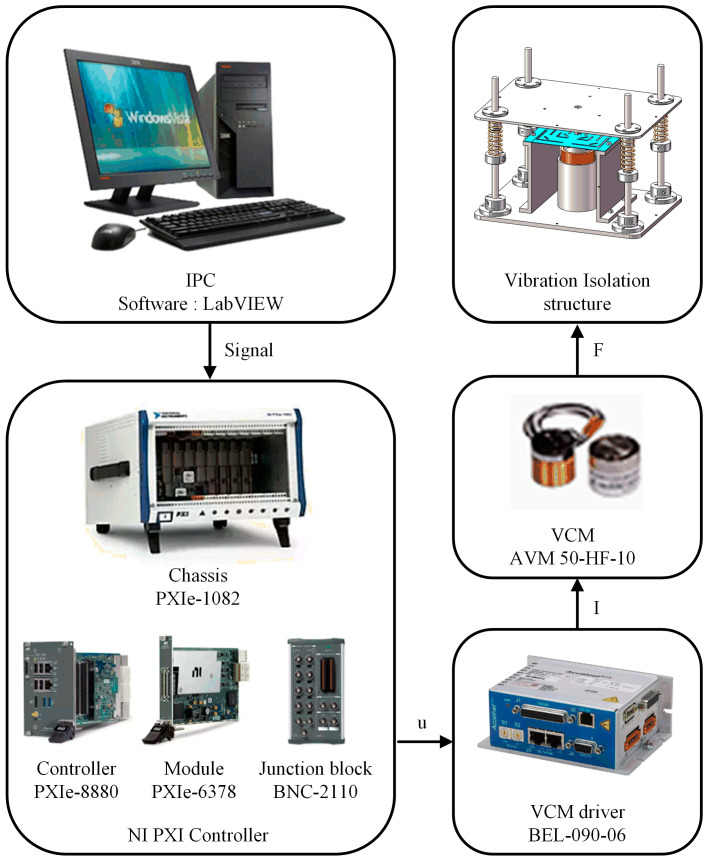
The flowchart of the position compensation loop signal.

**Figure 8 micromachines-14-01387-f008:**
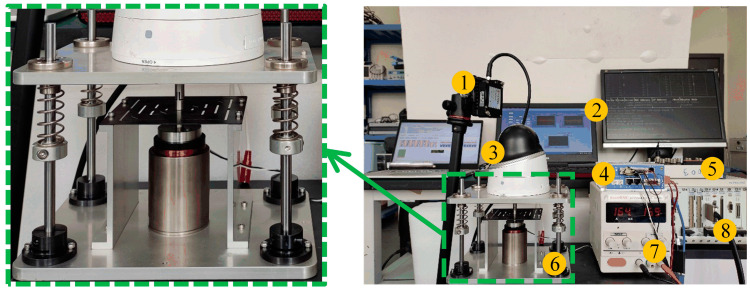
Experimental setup diagram of the position compensation loop testing. The labels are as follows: (1) doppler displacement sensors; (2) upper computer; (3) camera; (4) VCM driver; (5) junction block; (6) vibration isolation and compensation system (VICS); (7) DC power supply; (8) NI PXI controller.

**Figure 9 micromachines-14-01387-f009:**
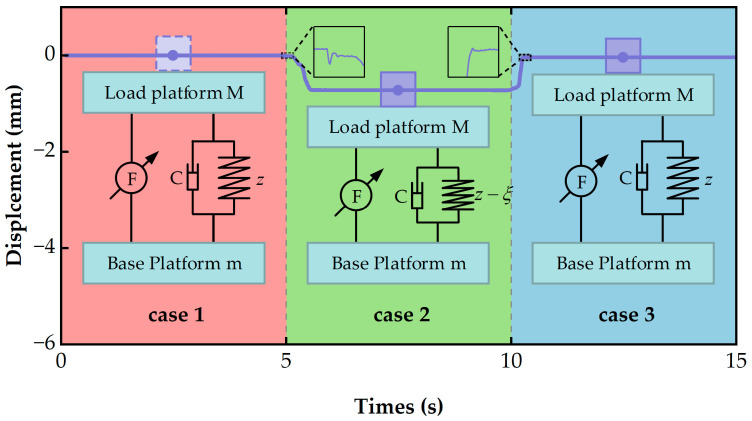
The results of the position compensation experiment: In Case 1, the camera head was not placed; in Case 2, the camera was placed; and in Case 3, the position compensation control force was applied.

**Figure 10 micromachines-14-01387-f010:**
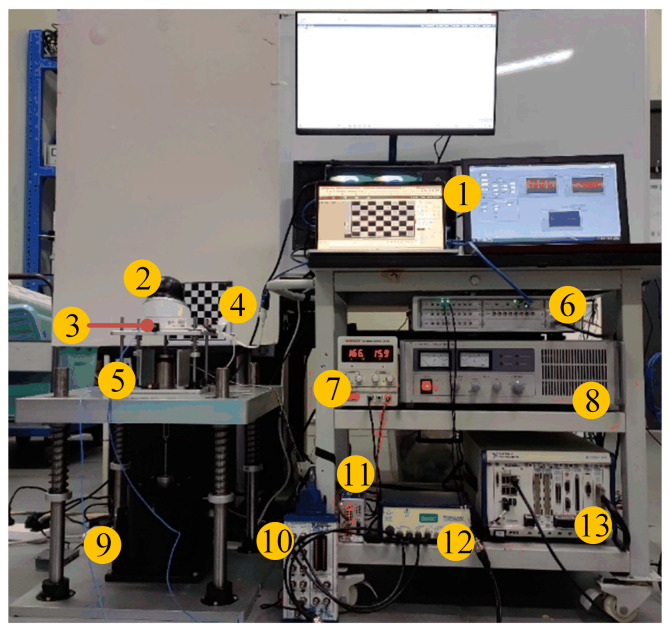
Experimental setup diagram of the vibration isolation performance test system. The labels are as follows: (1) upper computer; (2) camera; (3) acceleration sensor; (4) target; (5) VICS; (6) data acquisition system; (7) DC power supply; (8) power amplifier; (9) vibration exciter; (10) junction block; (11) VCM driver; (12) sensor signal conditioner; (13) NI PXI controller.

**Figure 11 micromachines-14-01387-f011:**
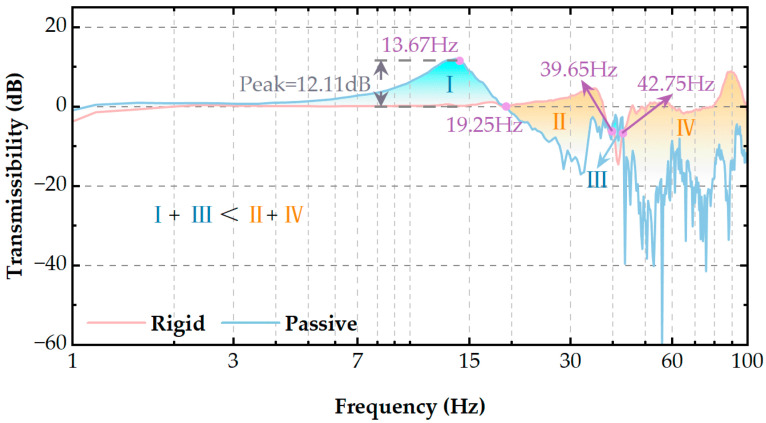
Comparison of the experimental results on the transmissibility of the rigid connection and the system with the addition of passive VICS.

**Figure 12 micromachines-14-01387-f012:**
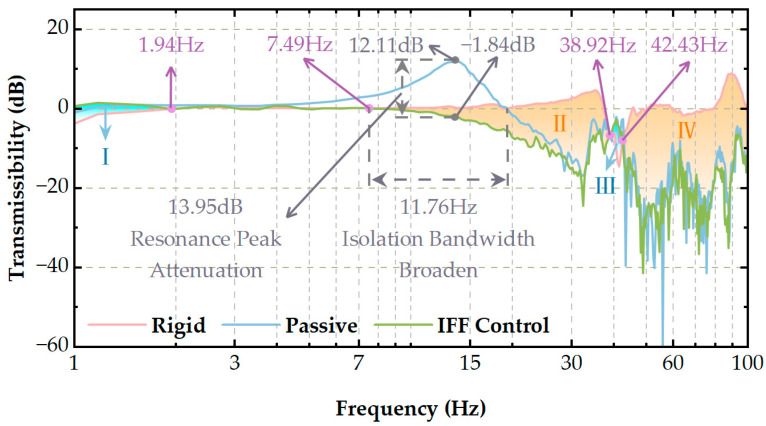
Comparison of the experimental results on the transmissibility of passive and integral force feedback (IFF) controls.

**Figure 13 micromachines-14-01387-f013:**
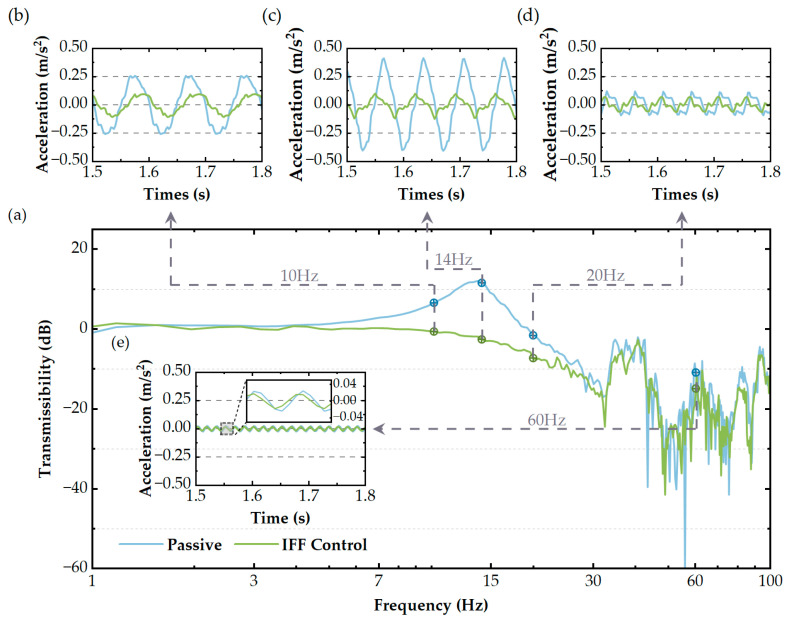
(**a**) In the range of 1–100 Hz, the transmissibility of the vibration isolator under white noise excitation is shown. The response in the time domain under periodic excitation with different frequencies: (**b**) 10 Hz; (**c**) 14 Hz; (**d**) 20 Hz; and (**e**) 60 Hz.

**Figure 14 micromachines-14-01387-f014:**
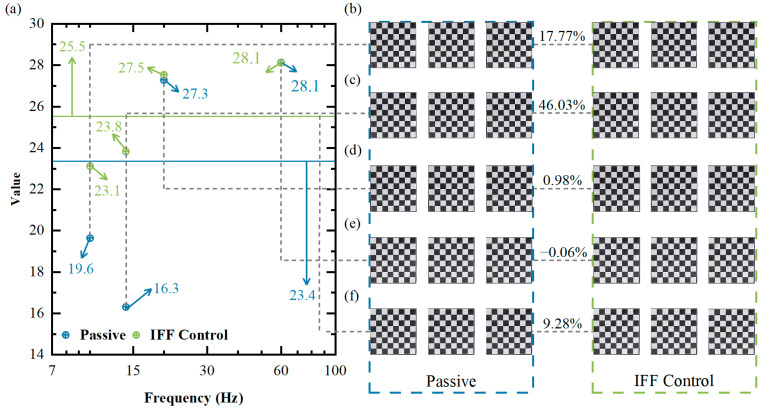
(**a**) Image performance index data; (**b**) 10 Hz mage data collected; (**c**) 14 Hz image data collected; (**d**) 40 Hz image data collected; (**e**) 60 Hz image data collected; (**f**) white noise image data collected.

**Table 1 micromachines-14-01387-t001:** Product parameters of HIKVCSION camera.

Parameter	Specifications
Model	DS-2DE2204IW-DE3/W/XM
Size	125 × 137.6 mm
Weight	650 g

**Table 2 micromachines-14-01387-t002:** The parameters of system identification.

Parameters	Values
Payload Mass *M* (Kg)	1.5
System stiffness $K_{e}$ (N/m)	11,066
System damping *C* (N S/m)	33

**Table 3 micromachines-14-01387-t003:** Material of the various components of the vibration isolation.

Name	Material
Load platform, base platform, flexible hinge, side panels, motor output shaft, fixed ring.	Al6061
Guide shaft support	S45C
Linear bearings, guide shaft	GCr15
Diaphragm spring, coil spring	65Mn

**Table 4 micromachines-14-01387-t004:** Model of the experimental equipment.

System Composition	Equipment Model
Data acquisition system	LMS SCADAS Mobile SCM205
Power amplifier	SA-PA050
Vibration exciter	SA-JZ020
Acceleration sensor	PCB-356A17
Sensor signal conditioner	PCB-482C05
NI controller	NI PXIe-1082, PXIe-8880, PXIe-6378, BNC-2110
DC power supply	MAISHENG MT-152D
VCM driver	BEL-090-06
VCM	AVM 50-HF-10
Upper computer	LMS Testlab software
	NI LabVIEW software
	HIKVCSION SADP

**Table 5 micromachines-14-01387-t005:** The frequency and time-domain signal under different conditions.

		Resonance Peak at Natural Frequency (13.67 Hz)	Acceleration Amplitude	Decrease
Random	Passive	12.11 dB	/	13.95 dB
	IFF	−1.84 dB		
10 Hz	Passive	/	±0.254 g	0.160 g
	IFF	/	±0.094 g	
14 Hz	Passive	/	±0.410 g	0.300 g
	IFF	/	±0.110 g	
20 Hz	Passive	/	±0.119 g	0.053 g
	IFF	/	±0.066 g	
60 Hz	Passive	/	±0.025 g	0.009 g
	IFF	/	±0.016 g	

**Table 6 micromachines-14-01387-t006:** Image performance index data recorded for different conditions.

Conditions	White Noise	10 Hz	14 Hz	20 Hz	60 Hz
Passive	23.36274	19.63535	16.31451	27.28037	28.13014
IFF	25.53195	23.12428	23.82355	27.54754	28.11190
Performance improvement	9.28%	17.77%	46.03%	0.98%	−0.06%

## Data Availability

The data supporting the findings of this paper is available from the corresponding authors on request.

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
