# Peer review of "An Active Vibration Isolation and Compensation System for Improving Optical Image Quality: Modeling and Experiment"

_micromachines, 2023, doi:10.3390/mi14071387_

Round 1

Reviewer 1 Report

This study presents a novel active control method aimed at enhancing the accuracy of optical image quality captured by cameras. Primarily centered around modeling and experimental research, the proposed method holds engineering application value. However, certain issues need to be addressed before the manuscript can be published.

(1) Section 2

The research on the new indicator introduced in this study requires more detailed analysis and discussion. The proposal of a new indicator necessitates a thorough comparison of its differences and advantages over traditional indicators. Therefore, it is crucial to use existing benchmark studies to demonstrate the effectiveness of the proposed indicator. Unfortunately, this study seems to have overlooked this aspect and needs to provide more substantial discussions on the rationality and scientific validity of the new indicator.

(2) Section 3

The experimental research detailed in this section appears to lack innovation in terms of the overall experimental design and the control methods employed. As a core component of this study, the experimental research needs to be better explained in terms of its innovation and contribution.

(3) Section 4.1

Accurate modeling of the mathematical model of the controlled system is crucial for effective active control. In this study, Equation (13) represents the mathematical model of the system. It would be helpful to explain how the parameters K and C in the model are calibrated, particularly the damping parameter C.

(4) Section 5.2

The experimental results demonstrate the effectiveness of the active control method employed in this study in controlling camera vibration. However, readers may have concerns about the stability of active control. While passive control may have stronger stability, it may be less effective in suppressing vibration in certain frequency ranges due to resonance. On the other hand, active control is more prone to instability. If instability occurs, what methods can be employed to avoid negative consequences such as control divergence?

(5) Table 7

The significance of the new indicator proposed in this study for improving and enhancing the effectiveness of active control needs to be clarified. Is it only used to indicate that active control is more effective than passive control, or does it have other implications for active control? Providing more detail on the importance of this new indicator would be helpful.

No additional comments

Reviewer 2 Report

This paper introduces an active vibration isolation system. The writing of the paper is standard and the data are abundant. But the writing is not easy to understand. After reading the paper carefully, I still did not find out how active vibration isolation is realized. I think the algorithm is just one part of active vibration isolation. What are the hardware, core principles and core devices? The above key contents should be clearly stated in the summary and conclusion. Some papers of the top journals should be added to the references, and some papers by authors from all over the world should be added. At present, most of the papers are by Chinese authors.

Round 2

Reviewer 1 Report

Thank you for the author's detailed reply and revision. The concerns have been well addressed, and the manuscript is recommended for acceptance.

Furthermore, an additional author has been included in the manuscript. The authors are required to specify the main contribution of the new author to this study.